# Effect of Antimicrobial Peptide Microcin J25 on Growth Performance, Immune Regulation, and Intestinal Microbiota in Broiler Chickens Challenged with *Escherichia coli* and *Salmonella*

**DOI:** 10.3390/ani10020345

**Published:** 2020-02-21

**Authors:** Gang Wang, Qinglong Song, Shuo Huang, Yuming Wang, Shuang Cai, Haitao Yu, Xiuliang Ding, Xiangfang Zeng, Jiang Zhang

**Affiliations:** 1State Key Laboratory of Animal Nutrition, College of Animal Science and Technology, China Agricultural University, Beijing 100193, China; crazygang@126.com (G.W.); songql@nferc.org (Q.S.); shuo0908@163.com (S.H.); wudixiaoming@163.com (Y.W.); c_caishuang@163.com (S.C.); 15600660793@163.com (H.Y.); ziyangzxf@163.com (X.Z.); 2National Feed Engineering Technology Research Center, Beijing 100193, China; sd_dingxl@Hotmail.com; 3Angelia Bio-Egineering Technology Co., Ltd., Chongqing 402460, China

**Keywords:** Microcin J25, antimicrobial peptide, *Salmonella*, *Escherichia coli*

## Abstract

**Simple Summary:**

Antimicrobial peptides are thought to be ideal candidates, owing to their antimicrobial properties, broad spectrum of activity, and low propensity for development of bacterial resistance. Microcin J25 (MccJ25) is an antimicrobial peptide produced by a fecal strain of *Escherichia coli* (*E. coli*) containing 21 Arbor Acres (AA) broilers, and has a strong inhibition on *E. coli* and *Salmonella*. This study was performed primarily to evaluate the effects of MccJ25 on growth performance and gut health in broilers challenged with *E. coli* and *Salmonella*, as a potential substitute for antibiotics. The results showed that MccJ25 promoted growth performance, improved intestinal morphology, and influenced fecal microbiota composition.

**Abstract:**

The purpose of this study was to investigate the effects of antimicrobial peptide microcin J25 (MccJ25) on growth performance, immune regulation, and intestinal microbiota in broilers. A total of 3120 one-day-old male Arbor Acres (AA) broilers were randomly allocated to five groups (12 replicates, 52 chickens per replicate). The treatments were control, challenge (0 mg/kg MccJ25), different dosages of antimicrobial peptide (AMP) (0.5 and 1mg/kg MccJ25), and antibiotic groups (20 mg/kg colistin sulfate). The MccJ25 groups increased the body weight gain (starter and overall) that was reduced in the challenge group. The overall (day 1 to day 42) feed-to-gain ratio (G:F) was significantly decreased in AMP groups compared with the challenge group. Birds fed AMP had a decreased population of total anaerobic bacteria (day 21 and day 42) and *E. coli* (day 21 and day 42) in feces, as well as a lower *Salmonella* infection rate (day 21 and day 42) compared with birds in the challenge group. The villus height of the duodenum, jejunum, and ileum, as well as the villus height/crypt depth of the duodenum and jejunum were greater in AMP groups than birds in the challenge group. Moreover, MccJ25 linearly improved the villus height of the duodenum and jejunum. The addition of MccJ25 decreased the concentration of TNF-α, IL-1β, and IL-6 compared with challenge group. At d 21, MccJ25 linearly reduced the level of IL-6. In conclusion, dietary supplemented MccJ25 effectively improved performance, systematic inflammation, and improved fecal microbiota composition of the broilers.

## 1. Introduction

*Escherichia coli* (*E. coli*) is very common in the natural environment. It can easily invade animals through drinking water, equipment, feces, etc. Avian colibacillosis is one of serious enteric diseases in poultry production, and it can cause decreased production performance, am increased mortality rate, and subsequent huge economic losses for the global poultry industry. The local or systemic diseases caused by avian *E. coli* mainly include sepsis, granuloma, peritonitis, osteomyelitis, salpingitis, and hemorrhagic enteritis. The incidence of colibacillosis is 1%–69%, the mortality rate is 3.8%–72.9%, and the lethality rate is 40.2%–90.3% [1]. Pullorum disease is a common multiple infectious disease caused by *Salmonella* pullorum, which seriously affects chicken survival rate. *E. coli* and *Salmonella* can increase the risk of the continuing association of poultry-borne transmission to humans.

At present, antibacterial drugs, such as colistin sulfate and tetracycline, are mainly used for preventing or controlling colibacillosis and salmonellosis in our country. Although antibiotics are effective means for preventing the occurrence of disease, the long-term use of antibiotics creates the problem of bacterial resistance and poses huge risks for human and animal health [2]. In recent years, many countries have severely restricted the use of antibiotics as health and growth promoters in livestock feed. Therefore, it is necessary to find alternative compounds from natural sources to control the incidence of colibacillosis and salmonellosis and reduce the loss of the poultry industry in the post-antibiotic era.

Antimicrobial peptides (AMPs) have been described as evolutionarily ancient weapons against microbial, viral, fungal, and protozoal activities [3]. AMPs are attractive candidates for new antimicrobial agents for specific applications, because of their natural antimicrobial properties and the low propensity for the development of bacterial resistance [4]. At present, there are more than 2000 AMPs originating from animals. However, the antibacterial spectrum, antibacterial activity, antibacterial mechanism, and even toxicity to normal cells of different antibacterial peptides have great differences [5]. Microcin J25 (MccJ25) is a new type of antimicrobial peptide containing 21 amino acids. It was discovered in the intestinal microbiota and has a strong inhibitory effect on gram-negative bacteria, such as *E. coli* and *Salmonella* [6]. MccJ25 essentially targets *Salmonella* and *E. coli*, with minimum inhibitory concentrations (MICs) in the nanomolar range (between 2 and 50 nM) [7]. In previous studies, supplementation with MccJ25 has had beneficial functions, not only decreasing *E. coli* in the gut but also maintaining microbiota homeostasis and reducing intestinal permeability; it also provokes anti-inflammatory responses in animal and cell models [7,8,9]. These features might make it useful for various industrial applications, such as in the poultry industry. Therefore, the objective of this study was to determine the efficacy of MccJ25 as a replacement for conventional antibiotics, in order to enhance the growth performance and intestinal heath of broilers when challenged with pathogens.

## 2. Materials and Methods

### 2.1. Animal Ethics Statement

All animal procedures used in this study were approved by the Animal Care and Use Committee at China Agricultural University (201605510410554).

### 2.2. Experimental Design and Management

This study was conducted using a total of 3120 one-day Arbor Acres (AA) male broilers (Beijing, China) in a completely randomized design for a 42-d period. Antibiotic-free and coccidiostat-free corn-soybean meal-based diets were formulated to meet or exceed nutrient requirements for broilers (NRC, 1994). The composition of the basal diet and nutrient levels for Arbor Acres broilers are presented in Table 1. Broilers were randomly assigned to 5 treatments groups, each with 12 replicates (52 chickens per replicate). From day 1 to day 7, all broilers fed basal diet, then changed to the experimental diet from d 8 to 14. Treatments consisted of a (1) control group, fed a basal diet, then groups (2), (3), (4) and (5) were challenged with *E. coli* AZ1 (1 × 10^6^ CFU/g) and *Salmonella* CVCC519 (1 × 10^6^ CFU/g) in drinking water, and fed a basal diet with 0 mg/kg Mcc J25 (2, challenge group), different AMP dosages of 0.5 mg/kg (3) and 1 mg/kg Mcc J25 (4), and an antibiotic, colistin sulfate at 20 mg/kg (5, Antibiotic group). On the 8th day, all the challenge was stopped in all groups and broilers were offered normal water.

### 2.3. Preparation of Bacterial Strain and MccJ25

The recombinant MccJ25 method was performed in our laboratory using a novel expression vector, as has been described previously [9]. Pure (99.95%) MccJ25 was dissolved in H_2_O to a final concentration of 12.8 mg/mL (*w*/*v*). The solution was centrifuged at 5000 rpm for 2 min, and the supernatant was filtered by a 0.22 μm filter and collected. *E. coli* AZ1 and *Salmonella* CVCC519 were cultured in Mueller–Hinton (MH) broth (Difco, Loughborough, UK) on a rotary at 120 rpm for 6 h until reaching the mid-logarithmic phase of growth. Then the cultures were centrifuged at 8500 rpm for 5 min, washed, and re-suspended in sterile Phosphate Buffered Saline (PBS) to obtain a final bacterial density of 10^6^ CFU/mL. The *E. coli* AZ1 is a multidrug resistant bacterium supplied by National Veterinary Drug Evaluation Center, the *Salmonella* CVCC519 was purchased from China Veterinary Microbial Species Conservation and Management Center (CVCC), and strain ATCC25922 was purchased from American Type Culture Collection (ATCC).

### 2.4. Determination of Minimum Inhibitory Concentration

The minimal inhibitory concentration of the MccJ25 was determined in sterilized 96-well microplates (Costar, Corning Inc., Corning, NY, USA) using microdilution assays, as has been described previously [10]. Briefly, MccJ25 was dissolved in H_2_O, and serial twofold dilutions were made in MH medium using 96-well microplates. Finally, the concentrations ranged from 0.0625 to 32 ug/mL. Fifty microliters of an overnight culture of *E. coli* AZ1 (5 × 10^5^ CFU/mL) and *Salmonella* CVCC519 (5 × 10^5^ CFU/mL) were inoculated into each well. The MIC was determined as the lowest concentration that inhibited visible growth of the bacteria after 16 h of incubation. In addition, the MIC determination contained both positive (media and bacteria) and negative (media only) controls.

### 2.5. Sample Collection and Preparation

Body weight and feed consumption for each replicate (52 chickens weighed together) were recorded for three periods—starter (days 1–21), finisher (days 22–42), and overall (days 1–42)—in order to calculate the average daily gain (ADG), and G:F, as well as mortality.

At day 21 and day 42, 5 g of fresh feces from each replication were collected with an iron tray six times at different times of the day, and the feces were mixed. Twelve samples of each group were used for detecting the counts of *E. coli*, Lactic acid bacteria (LAB), and *Bifidobacterium*.

At day 42, 12 birds per group were randomly selected, euthanized, and sampled. Tissue samples from the duodenum, jejunum, and ileum were washed in PBS and immediately fixed in 4% paraformaldehyde for histological examination. For cytokine analysis, 10 mL wing venous blood was collected from each broiler and centrifuged at 3000 r/min for 10 min at room temperature, and then stored frozen at −80 °C.

### 2.6. E. coli, Lactid Acid Bacteria, Bifidobacterium Count, and Salmonella Infection Rate in Feces

Approximately 0.5 g feces were diluted with sterile, buffered peptone water (CM201, Land Bridge Technology Ltd., Beijing, China) to an initial 10^−1^ dilution, and the tube contents were homogenized separately using a Heidolph Diax 600 homogenizer (Heidoloh, Schwabach, Germany) for 30 s. Then the solutions were serially diluted from 10^−1^ to 10^−7^ in 0.85% sterile saline solution, and 100 μL of each diluted sample was subsequently plated on selective agar plates for bacteria counting.

*E. coli* were cultured using MacConkey’s agar (MAC) under anaerobic conditions at 37 °C for 24 h. LAB counts were determined using Man, Rogosa, Sharpe (MRS) agar and incubated at 37 °C for 24 h. *Bifidobacterium* colonies were counted on *Bifidobacterium* Bismuth Sulfite (BS) agar after incubation under an anaerobic cabinet at 37 °C for 16 h. The *Salmonella* infection rate was determined using enzyme-linked immunosorbent assay (ELISA) kits (Nanjing Jiancheng Bioengineering Institute, Nanjing, Jiangsu, China), based on standard procedures described by the manufacture.

### 2.7. Small Intestine Morphology

Gut histomorphology analysis was performed as has been previously described [11]. Paraformaldehyde-fixed intestinal samples were dehydrated and embedded in paraffin. Transverse sections were cut at 5 μm and stained with hematoxylin and eosin for histological analysis and measurement. Villus height and crypt depth were measured under an Olympus CK 40 microscope (Olympus Optical Company, Shenzhen, China) with 40 *×* 10 magnification.

### 2.8. Determination of Immunomodulatory factor Concentrations in Serum

The concentrations of IL-1β, IL-6, IgA, IgG, and tumor necrosis factor-α in the serum were measured using commercially available chicken enzyme-linked immunosorbent assay kits (Nanjing Jiancheng Bioengineering Co., Ltd., Nanjing, China). Data were acquired by a microplate reader using MPM 6.1 Software (Bio-Rad Laboratories, Hercules, CA, USA).

### 2.9. Statistical Analysis

All data were analyzed by ANOVA using the GLM procedure of SAS (version 9.4, SAS Institute, Inc., Cary, NC, USA). All means are presented as mean ± standard error of the mean (SEM). A Student-Newman-Keuls (SNK) multiple comparison test was used to determine the statistical difference between treatments. A *p*-value of less than 0.05 was considered statistically significant.

## 3. Results

### 3.1. Minimum Inhibitory Concentration (MIC) of MccJ25 Against E. coli AZ1 and Salmonella CVCC519

The inhibitory effects on growth of *E. coli* AZ1 and *Salmonella* CVCC519 at different concentrations are shown in Table 2. The MICs of MccJ25 against *E. coli* AZ1 and *Salmonella* CVCC519 were 1.0 μg/mL and 0.5 μg/mL, respectively. The MIC against positive control *E. coli* ATCC25922 was 1 μg/mL, and no growth was observed in the negative control (media only). These results suggest that MccJ25 has a good inhibitory effect on *E. coli* and *Salmonella*.

### 3.2. Growth Performance

The body weight (BW) gain, G:F, and mortality of the broilers in each treatment during various phases of production are summarized in Table 3. *E. coli* AZ1 and *Salmonella* CVCC519 challenge resulted in a significant decrease in BW (starter and overall), as well as a remarkable deterioration in G:F (starter) and mortality (starter, overall, and days 1–14) compared with the control treatment (*p* < 0.05). The addition of 0.5 and 1 mg/kg MccJ25 can significantly inhibit the effects of *E. coli* AZ1 and *Salmonella* CVCC519 on the growth performance of broilers, reducing the G:F on day 42 and mortality during the whole trial period (*p* < 0.05) and increasing the BW (during the starter and overall phase *p* < 0.05). There was no difference in BW, G:F, and mortality between the two MccJ25 treatments and the antibiotic group. However, *E. coli* AZ1 and *Salmonella* CVCC519 did not influence feed intake at day 21 and day 42. There was no difference between all the groups.

### 3.3. Count of Feces Bacteria and Salmonella Infection Rate

Results for microbial populations in feces and the *Salmonella* infection rate are shown in Figure 1. There was a significant increase of *E. coli*, total aerobic bacteria counts in the feces, and *Salmonella* infection rate at days 21 and 42, following seven days of *E. coli* AZ1 and *Salmonella* CVCC519 challenge, compared with the control group (*p* < 0.05). On day 21, the numbers of LAB and *Bifidobacterium* in broilers fed MccJ25 and antibiotics were greater than those in the challenge group (*p* < 0.05). A significant reduction in *E. coli* count and total aerobic bacteria in feces was found in the broilers fed MccJ25 and antibiotics (*p* < 0.05), but the amount was higher than in the control treatment (*p* < 0.05) at day 21 and day 42. On day 42, birds fed MccJ25 had a higher number of *Bifidobacterium* (*p* < 0.05) and lower *E. coli* counts (*p* < 0.05) in feces. The addition of MccJ25 significantly reduced *Salmonella* infection rate at day 42 (*p* < 0.05) compared with the challenge group, but no influence at day 21 (*p* > 0.05). Moreover, compared with the antibiotic group, the diet containing MccJ25 sharply decreased the *Salmonella* infection rate (*p* < 0.05) at day 42.

### 3.4. Intestinal Morphology

Compared the control treatment, *E. coli* AZ1 and *Salmonella* CVCC519 challenge severely damaged the small intestinal morphology, and significantly decreased the villus height, crypt depth, and villus height/crypt depth ratio of the duodenum and ileum, as well as the villus height and villus height/crypt depth ratio in jejunum (*p* < 0.05), whereas there was no effect on crypt depth in the jejunum (*p* > 0.05) (Table 4).The MccJ25 addition significantly increased the villus height and villus height/crypt depth in the duodenum and jejunum compared with challenge group (*p* < 0.05). Birds fed 0.5 and 1 mg/kg MccJ25 had a higher villus height and villus height/crypt depth ratio in jejunum and villus height in duodenum than those in the antibiotic group (*p* < 0.05), and 1 mg/kg MccJ25 had a better inhibition of intestinal morphological damage. In the ileum, there was no remarkable difference existed between MccJ25 treatments and the antibiotic group (*p* > 0.05). However, there was no effect on crypt depth in the small intestine in birds fed MccJ25 and antibiotic (*p* > 0.05).

Similar results were found in the intestinal mucosal morphology. Hematoxylin-eosin staining of the intestine showed broader damage and more severe inflammatory infiltration after *E. coli* AZ1 and *Salmonella* CVCC519 challenged in the duodenum and ileum (Figure 2). While MccJ25 and antibiotic therapy can alleviate mucosal damage (Figure 2). There were no significant differences in the intestinal mucosal morphology of jejunum (Figure 2).

### 3.5. Immune Factor Concentration in Serum

The results of the effects of MccJ25 on immunomodulatory factor concentration in the serum are shown in the Figure 3. *E. coli* AZ1 and *Salmonella* CVCC519 challenge had significantly increased the secretion of pro-inflammatory factors IL-1β, TNF-α, and IL-6 compared with the other four groups (*p* < 0.05). The levels of TNF-α, IL-1β, and IL-6 in the serum at day 21 in the 0.5 and 1.0 mg/kg MccJ25 groups were lower than in the control group, and the 1.0 mg/kg MccJ25 had a lower level of IL-6 compared with the 0.5 mg/kg. At day 42, the IL-1β and IL-6 concentrations in the serum were similar to the control group at day 42 (*p* < 0.05). However, the cytokines in broilers fed MccJ25 and antibiotic diets were similar. The concentration of immunoglobulin A (IgA) and immunoglobulin G (IgG) in the serum did not change in all diets (*p* > 0.05).

## 4. Discussion

Several studies have demonstrated the harmful effects of *E. coli*- and *Salmonella*-contaminated diets on broiler performance. Emami et al. showed that the supplementation of *E. coli K88* into a broiler’s diet led to lower body weight and smaller body mass [12]. Jazi et al. and Vandeplas et al. reported that the *Salmonella Typhimurium* challenge decreased feed intake and body weight gain, and increased feed conversion ratios in broiler chickens [13,14]. The data in the present study also demonstrates that *E. coli* and *Salmonella* severely affect the growth and feed intake of birds and increase the efficiency of gain. Addition of MccJ25 in the diet can greatly improve the growth performance and reduce the mortality of birds. This indicates that MccJ25 has a certain resistance to *E. coli* and *Salmonella*. Previous studies have also shown the benefits of AMPs on the growth performance of swine and poultry. Wang et al. reported that lactoferrin isolated from milk could increase the average daily gain and the efficiency of gain of weaning piglets by 41.80% and 17.20%, respectively [15]. Dietary supplementation with nisin (E234) has the potential to improve growth performance and nutrient retention in broilers [16].

Antibiotics improve growth performance, mainly because they inhibit the colonization of pathogenic bacteria in the intestine in broiler. The results of minimal inhibitory concentration in the in vitro experiment showed that MccJ25 could inhibit the growth of *E. coli* and *Salmonella*. Meanwhile, the counts of *E. coli* and *Salmonella* in the feces were reduced after the supplementation with MccJ25, which indicates that MccJ25 has an antibacterial effect similar to that of antibiotics. Additionally, the present study shows that MccJ25 increases the numbers of *Bifidobacterium* and lactic acid bacteria. Probiotics also have the potential to prevent the colonization of pathogens by competing for nutrients and epithelial binding sites, as well as through the production of antimicrobial factors, such as lactic acid and bacteriocins. Similar to our results, Ohh et al. reported that dietary supplementation of AMPs isolated from *S. tuberosum* had fewer coliforms in the excreta and cecal digesta in broilers [17]. Several studies in swine showed that many AMPs, such as lactoferrin (LF) and AMP A5 (A3) decreased the counts of *E. coli* and *Salmonella*, and increased the *Lactobacillus* counts in the small intestine of swine [8,18,19,20]. It is understood that AMPs beneficially affect animal health through improving intestinal health and creating gut micro-ecological conditions that suppress harmful microorganisms and promote beneficial microorganisms.

As we know, the ability for nutrient absorption, especially in the small intestine, has a great impact on animal growth. The absorptive surface in the small intestine is enormously enlarged by folds and by villi [21]. A shortening of the villus would result in poor nutrient absorption, diarrhea, weak disease resistance, and lower performance [22]. *E. coli* and *Salmonella* infection would lead to villous atrophy and intestinal morphology disruption [12,13]. In the present study, apparent alterations in the villus–crypt structure in the duodenum and jejunum, such as higher villus height and villus height/crypt depth, were observed in broilers after the supplementation of MccJ25. In line with results of the present study, Wu et al. reported that a higher villus height/crypt depth in the jejunum and ileum, as well as higher villus height, were observed when fed the AMP cecropin AD after an *E. coli* challenge [23]. Similar results were also observed in other AMPs [18,24]. Increased villus height and villus height/crypt depth are correlated with epithelial turnover and activation of cell mitosis [25,26]. This may explain how MccJ25 can improve the growth performance after challenge in the boiler.

Intestinal dysfunction destroys the structural integrity between enterocytes and lamina propria, leading to strong inflammatory reactions in birds, then impaired nutrient utilization, and finally suppressed broiler performance. The present study shows that *E. coli* and *Salmonella* infection lead to broader damage and more severe inflammatory infiltration in the small intestine, while MccJ25 can alleviate mucosal damage. Cytokines play an important role in immunoregulation, and any imbalance in cytokine production or deregulation would lead to various pathological disorders [27]. MccJ25 dramatically reduced the secretion of proinflammatory factors IL-1β, IL-6, and TNF-α in the serum. This means that systemic inflammation was restored. Similarly, Hing et al. illustrated that cathelicidin decreases toxin A-induced TNF-α expression via the inhibition of nuclear factor-kappa B (NF-κB) activity, which relieves the *Clostridioides difficile* (*C. diffcile*)-mediated intestinal inflammation [28].

## 5. Conclusions

In brief, MccJ25 promoted growth performance, improved the intestinal morphology, decreased the inflammation and influenced fecal microbiota composition in broiler under the challenge of *E. coli* and *Salmonella*.

## Figures and Tables

**Figure 1 animals-10-00345-f001:**
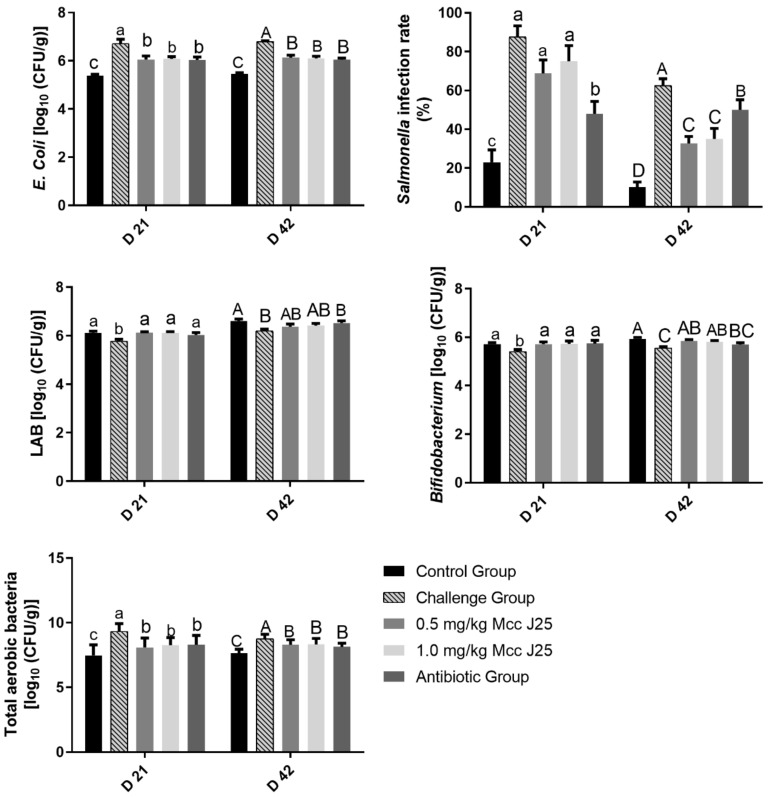
Effects of MccJ25 on *Escherichia coli* (*E. Coli*), lactic acid bacteria (LAB), *Bifidobacterium* counts (log 10 cfu/g), and total aerobic bacteria in the feces, as well as the *Salmonella* infection rate in broilers on days 21 and 42. Bars represent means ± SEM for 12 broilers per treatment. Within the same day, bars with different letters differ significantly (*p* < 0.05).

**Figure 2 animals-10-00345-f002:**
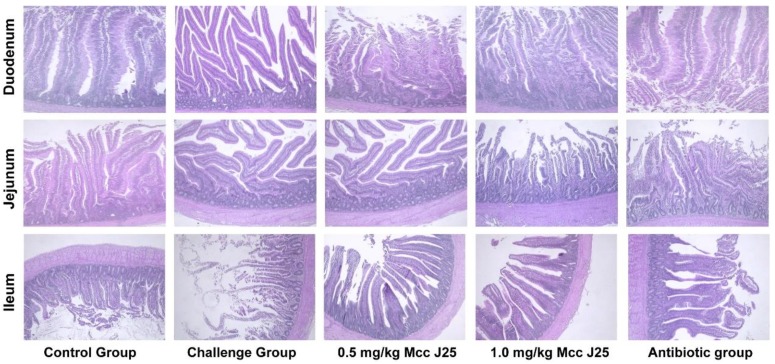
Effect of MccJ25 on intestinal mucosal morphology of duodenum, jejunum, and ileum in broilers. Scale bars, 100 μm.

**Figure 3 animals-10-00345-f003:**
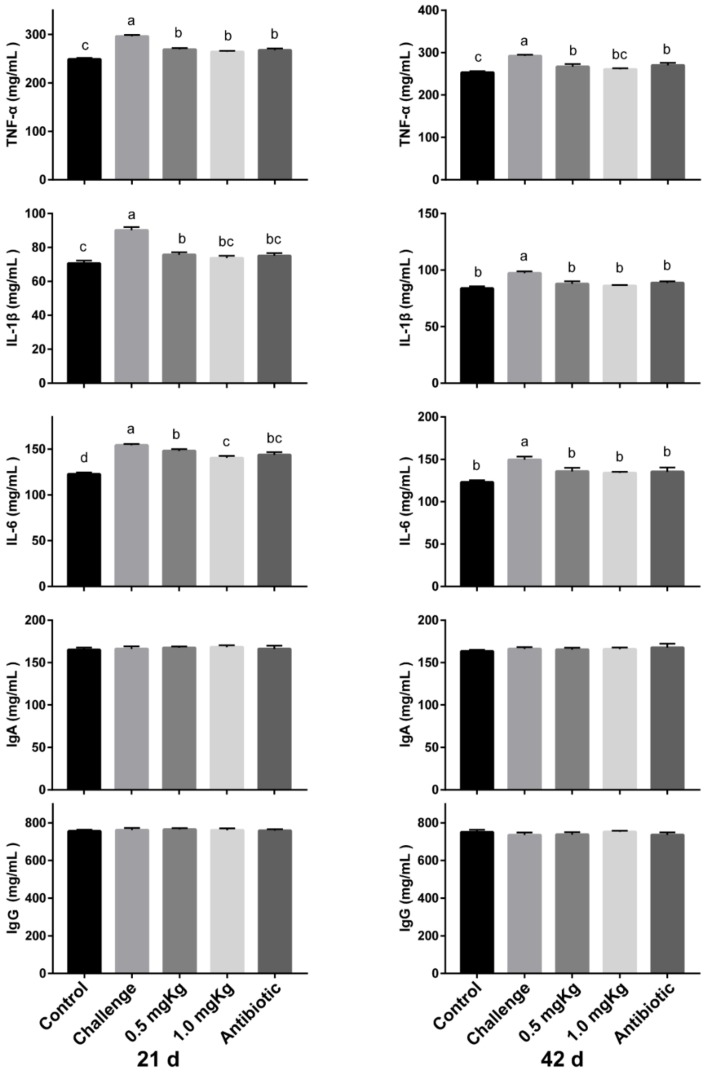
Effects of MccJ25 on immunomodulatory factors concentration in the serum on day 21 and day 42. Bars represent means ± SEM for 12 broilers per treatment. Bars with different letters differ significantly (*p* < 0.05).

**Table 1 animals-10-00345-t001:** Composition of basal diet and nutrient levels (as-fed basis).

Item	(1–21 d)	(21–42 d)
Ingredients (%)		
Corn	55.97	61.37
Soybean meal	37.63	31.88
Soybean oil	2.37	3.13
Limestone	1.27	1.21
NaCl	0.35	0.35
CaHPO_4_	1.75	1.49
DL-Methionine	0.2	0.12
HCl L-Lys·HCl	0.02	0.01
Antioxygen	0.02	0.02
Vitamin premix ^1^	0.02	0.02
Choline chloride ^1^	0.2	0.2
Trace mineral premix ^2^	0.2	0.2
Total	100	100
Analyzed nutritional value (%) ^3^		
Metabolic energy/(MJ/kg)	11.76	12.22
Crude protein	21.36	19.87
Crude fat	4.83	5.02
Crude fiber	3.46	3.09
Ca	1.01	0.91
Total phosphorus	0.60	0.54
Lys	1.14	1.02
Met	0.50	0.41

^1^ The vitamin premix provided the following per kg of feed: vitamin A, 9500 IU; vitamin D_3_, 2500 IU; vitamin K_3_, 2.65 mg; vitamin B_12_, 25 μg; vitamin B_2_, 6 mg; vitamin E, 30 IU; biotin, 0.0325 mg; folic acid, 1.25 mg; pantothenic acid, 12 mg; nicotinic acid, 50 mg; ^2^ The mineral premix provided the following per kg of feed: Cu, 8 mg; Zn, 75 mg; Fe, 80 mg; Mn, 100 mg; Se, 0.15 mg; and I, 0.35 mg; ^3^—Values are the means of a chemical analysis conducted in duplicate.

**Table 2 animals-10-00345-t002:** Minimum inhibitory concentration (MIC) of microcin J25 (MccJ25) against *E. coli* AZ1 and *Salmonella* CVCC519.

MccJ25 μg/mL	0	0.0625	0.125	0.25	0.5	1	2	4	8	16	32
*E. coli* AZ1	+	+	+	+	−	−	−	−	−	−	−
*Salmonella* CVCC519	+	+	+	+	−	−	−	−	−	−	−
*E. coli* ATCC25922	+	+	+	+	+	−	−	−	−	−	−

+ mean grew, − mean did not grow.

**Table 3 animals-10-00345-t003:** Effects of MccJ25 on performance of broilers under *E. coli.* and *Salmonella* infection ^1^.

Item	Treatment					SEM	*p*-Value
	Control Group	Challenge Group	0.5 mg/kg MccJ25	1.0 mg/kg MccJ25	Antibiotic Group		
BW (g)							
days 1–21	872 ^a^	823.25 ^d^	857.67 ^ab^	840.24 ^c^	846.18 ^bc^	5.32	<0.01
days 22–42	1619.09	1576.98	1604.12	1623.53	1603.51	14.37	0.19
days 1–42	2491.1 ^a^	2400.2 ^c^	2461.8 ^ab^	2463.8 ^ab^	2449.7 ^bc^	14.98	<0.01
Feed-to-Gain ratio (G:F)							
days 1–21	1.36 ^b^	1.42 ^a^	1.39 ^ab^	1.4 ^ab^	1.37 ^b^	0.014	0.04
days 1–42	1.76 ^c^	1.9 ^a^	1.82 ^b^	1.83 ^b^	1.84 ^b^	0.01	<0.01
Feed Intake (kg)							
days 1–21	1.26	1.24	1.26	1.25	1.23	0.02	0.63
days 1–42	4.47	4.66	4.57	4.60	4.60	1.49	0.45
Mortality (%)							
days 1–14	0.32 ^b^	2.72 ^a^	1.12 ^b^	1.28 ^b^	1.16 ^b^	0.32	<0.01
days 1–21	0.48 ^c^	4.01 ^a^	2.08 ^b^	2.08 ^b^	1.71 ^b^	0.42	<0.01
days 1–42	0.52 ^c^	5.03 ^a^	2.78 ^b^	2.43 ^b^	2.43 ^b^	0.41	<0.01

^1^ Each mean represents 12 replicates; ^a,b,c,d^ Means in the same row with different superscripts differ (*p* < 0.05).

**Table 4 animals-10-00345-t004:** Effects of MccJ25 on villus height, crypt depth, and villus height to crypt depth of duodenum, jejunum and ileum in broilers ^1^.

Item	Experimental Treatments					SEM	*p*-Value
	Control Group	Challenge Group	0.5 mg/kg MccJ25	1.0 mg/kg MccJ25	Antibiotic Group		
Duodenum							
Villus height (μm)	754.82 ^a^	636.27 ^e^	686.04 ^c^	695.05 ^b^	672.18 ^d^	2.68	<0.01
Crypt depth (μm)	101.21 ^a^	90.56 ^b^	90.2 ^b^	91.42 ^b^	90.12 ^b^	0.64	<0.01
Villus height/crypt depth	7.46 ^a^	7.03 ^b^	7.61 ^a^	7.61 ^a^	7.46 ^a^	0.06	<0.01
Jejunum							
Villus height (μm)	322.27 ^bc^	303.57 ^d^	325.15 ^b^	331.47 ^a^	318.78 ^c^	1.92	<0.01
Crypt depth (μm)	94.24	97.94	96.66	96.37	97.29	1.30	0.33
Villus height/crypt depth	3.42 ^ab^	3.11 ^c^	3.37 ^ab^	3.44 ^a^	3.28 ^b^	0.05	<0.01
Ileum							
Villus height (μm)	252.94 ^a^	242 ^b^	236.19 ^b^	242.56 ^b^	245.63 ^ab^	3.10	<0.01
Crypt depth (μm)	63.07 ^b^	75.11 ^a^	60.72 ^b^	60.53 ^b^	60.85 ^b^	0.90	<0.01
Villus height/crypt depth	4.02 ^a^	3.23 ^b^	3.9 ^a^	4.02 ^a^	4.04 ^a^	0.07	<0.01

^1^ Each mean represents 12 replicates; ^a,b,c,d,e^ Means in the same row with different superscripts differ (*p* < 0.05).

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
