# Peer review of "Effect of Antimicrobial Peptide Microcin J25 on Growth Performance, Immune Regulation, and Intestinal Microbiota in Broiler Chickens Challenged with Escherichia coli and Salmonella"

_animals, 2020, doi:10.3390/ani10020345_

Round 1

Reviewer 1 Report

Continued research on alternatives to AGPs is necessary, especially for countries that still use them for production purposes. However, a clear background on the alternative, supported by literature is necessary, even if it focus on its biochemical properties and physiological effects. The Introduction lacks the background and justification for investigating the potential of this product.

Without giving much detail, the description of treatments in the Abstract as in the Materials and Methods should be clear and precise. This is lacking.

The methodology followed in this study is similar to many other studies where birds are challenged to test an antimicrobial additives. But the intentional exposure of birds to pathogens to weaken their immunity raises ethical concerns. In essence, birds reared in an environment with optimal biosecurity are unlikely to suffer from any pathogenic diseases.

Author Response

Response to Reviewer 1 Comments

Major points:

Comments to the Author: Continued research on alternatives to AGPs is necessary, especially for countries that still use them for production purposes. However, a clear background on the alternative, supported by literature is necessary, even if it focus on its biochemical properties and physiological effects. The Introduction lacks the background and justification for investigating the potential of this product.

Response 1: Thanks for this comment. We have supplemented the detail in the background including previous research about the AMPs and the feasibility of the trial.

Comments to the Author: Without giving much detail, the description of treatments in the Abstract as in the Materials and Methods should be clear and precise. This is lacking.

Response 2: Thanks for the comment. We have added the detail in the Materials and Methods in the revised manuscript.

Comments to the Author: The methodology followed in this study is similar to many other studies where birds are challenged to test an antimicrobial additive. But the intentional exposure of birds to pathogens to weaken their immunity raises ethical concerns. In essence, birds reared in an environment with optimal biosecurity are unlikely to suffer from any pathogenic diseases.

Response 3: It is really true as reviewer said that intentional exposure of birds to pathogens to weaken their immunity and it is a bit extreme in the production. But the purpose of our trial is to study the potential of antimicrobial peptides to replace antibiotics. E.coli and Salmonella are common pathogens in the poultry industry and they brought a huge economic lose every year.

We are very sorry for these points because our incorrect and un-rigorous writing. And we have changed it in the test. Special thanks to you for your good comments.

We tried our best to improve the manuscript and made some changes in the manuscript. These changes will not influence the content and framework of the paper. And here we did not list the changes but marked in yellow in revised paper.

We appreciate for reviewer’s warm work earnestly, and hope that the correction will meet with approval.

Once again, thank you very much for your comments and suggestions.

Reviewer 2 Report

L18: The aim and genral objective should be rephased and adapted to become clear and fit to with the work therefore developed

L25: What are the different dosages of AMP used ?

Conclusions at the end of the abstract: Indicate the main conclusions or interpretations at the end of the abstract

L68-70:  Please rephrase this part of the introduction with clearer hypothesis and objectives of the study.

L72: Please provide the approval number of the Ethics Committee for this research.

L77: were randomly assigned, distributed…

L79-83: Please make the description of treatments clearer. As the sentence is written, it is not clear if the AMP groups are under the same challenge or not.

L78: starting from “at first 7days”: please use a passive form

L83 starting from after 7 days: please rephrase the sentence

L85: Why authors used requirements of poultry (1994).  Levels of energy and crude proteins are below those mentioned in a more recent Arbor Acres broilers nutrition specification. This may interfere with the effect of challenge on productive performance of the chickens.

The methodology lacks the challenge procedure

L117-119: please rephrase this part of the text.

L137: Please use either “analyses were” or “analysis was”

Table 2. Please complete the title.

L171: please put did not influence instead of did not influenced.

Table3. It seems that there is a mistake in the control group who ate 7.81 to reach 2491.1 g at the end of the experiment. How did authors get the value of 1.76 for the feed efficiency? Also it is so strange that p-value of feed intake 1-42d is 0.45 with a treatment whose intake was 7.81 kg vs 4.60 for the rest.

Figure1: missed

Table4. Please review the superscripts for the for the villus height of the duodenum villus height. Authors put d for 672.18 and c for 636.27.

L204-207: The meaning is not clear.

L255-258: Please rephrase the sentence starting from in the present study until jejunum.

L259: The sentence preceding the citation 6 is a redundancy of what was said in lines 251-254

General comments:

-The whole part of discussion should be rewritten. The points to be discussed are effects of the Mcc J25 on Intestinal Morphology, bacteria count and immune response and to find the relationship between these parameters with the intention to support or to explain the improvement of productive performance.

- "References" is for improvement according to the journal guidelines. For example reference 30 the year is at the and after the volume and pages while it should be before, reference 12 lacks the number of pages… 

Author Response

Response to Reviewer 2 Comments( the attachment is the revised manuscript)

Major points:

Comments to the Author: L18: The aim and general objective should be rephased and adapted to become clear and fit to with the work therefore developed.

Response 1: Thanks for this comment. We have revised the aim and general objective in the “Simple Summary” part.

Comments to the Author: L25: What are the different dosages of AMP used?

Conclusions at the end of the abstract: Indicate the main conclusions or interpretations at the end of the abstract

Response 2: Thanks for the comments. We have revised in the revised manuscript.

Comments to the Author: L68-70:  Please rephrase this part of the introduction with clearer hypothesis and objectives of the study.
L72: Please provide the approval number of the Ethics Committee for this research.
L77: were randomly assigned, distributed…
L79-83: Please make the description of treatments clearer. As the sentence is written, it is not clear if the AMP groups are under the same challenge or not.
L78: starting from “at first 7days”: please use a passive form
L83 starting from after 7 days: please rephrase the sentence

Response 3: Thanks for these valuable suggestions. We have made the changes in the revised paper according to the comments.

Comments to the Author: L85: Why authors used requirements of poultry (1994). Levels of energy and crude proteins are below those mentioned in a more recent Arbor Acres broilers nutrition specification. This may interfere with the effect of challenge on productive performance of the chickens.

The methodology lacks the challenge procedure

Response 4: It is true that the new broilers nutrition standard is more suitable. But in our previous research, we did not find NRC 1994 will impact the production performance in broiler. In further trials, we will consider using a more recent standard. Thanks for the suggestion.

The challenge procedure was in the part of “Experimental Design and Management”. We did not list it separately.

Comments to the Author: L117-119: please rephrase this part of the text.
L137: Please use either “analyses were” or “analysis was”
Table 2. Please complete the title.
L171: please put did not influence instead of did not influenced.

Response 5: We have made the change in the revised paper by these comments.

Comments to the Author: Table3. It seems that there is a mistake in the control group who ate 7.81 to reach 2491.1 g at the end of the experiment. How did authors get the value of 1.76 for the feed efficiency? Also, it is so strange that p-value of feed intake 1-42d is 0.45 with a treatment whose intake was 7.81 kg vs 4.60 for the rest.

Response 6: We are very sorry for this mistake. We have checked the raw data. The food intake in d1- 42 in the Control group is 4.47, not 7.81. There were no differences among the groups.

Comments to the Author: Figure1: missed

Response 7: We are sorry for our mistake. We have added the Figure1 in the revised manuscript.

Comments to the Author: Table4. Please review the superscripts for the for the villus height of the duodenum villus height. Authors put d for 672.18 and c for 636.27.

Response 8: We have checked the superscripts carefully. The superscript in the 636.27 is e, not c.

Comments to the Author: L204-207: The meaning is not clear.

Response 9: We have made some changes in the revised paper.

Comments to the Author: L255-258: Please rephrase the sentence starting from in the present study until jejunum.

Response 10: We have rewritten this sentence.

Comments to the Author: L259: The sentence preceding the citation 6 is a redundancy of what was said in lines 251-254 General comments:

Response 11: We have rewritten this paragraph.

General comments:

Comments to the Author: The whole part of discussion should be rewritten. The points to be discussed are effects of the Mcc J25 on Intestinal Morphology, bacteria count and immune response and to find the relationship between these parameters with the intention to support or to explain the improvement of productive performance.

Response 1: Thanks for the suggestion. We have rewritten the discussion according to the comment.

Comments to the Author: "References" is for improvement according to the journal guidelines. For example reference 30 the year is at the and after the volume and pages while it should be before, reference 12 lacks the number of pages…

Response 2: We have renumbered the references and checked the format carefully according to the journal guidelines.

We are very sorry for these points because our incorrect and un-rigorous writing. And we have changed it in the test. Special thanks to you for your good comments.

We tried our best to improve the manuscript and made some changes in the manuscript. These changes will not influence the content and framework of the paper. And here we did not list the changes but marked in yellow in revised paper.

We appreciate for reviewer’s warm work earnestly, and hope that the correction will meet with approval.

Once again, thank you very much for your comments and suggestions.

Round 2

Reviewer 1 Report

This is a well-thought study with originality and merit. However, there are several areas with inconsistencies that need clarity and correction. Please see the document for comments.

The treatment description still needs further attention. Treatment 2 cannot be 'challenge group' if 3, 4, and 5 were also challenged, then treated with additives. 

Were response from the AMP treatments, that is, 3 and 4 similar, seeing that they are presented as one, even in the conclusion. Were there no response differences with the varying dosage levels? If that is the case, great, but the conclusion should state that precisely. 

Author Response

Response to Reviewer 1 Comments (the attachment was the revised manuscript )

1. Comments to the Author: The treatment description still needs further attention. Treatment 2 cannot be 'challenge group' if 3, 4, and 5 were also challenged, then treated with additives.

Response1: Thanks for the suggestion.we have changed the 'challenge group' to "ES group" in the reviesd manusript. 

2. Comments to the Author: Were response from the AMP treatments, that is, 3 and 4 similar, seeing that they are presented as one, even in the conclusion. Were there no response differences with the varying dosage levels? If that is the case, great, but the conclusion should state that precisely. 

Response2: Thanks for the suggestion. we have changed it in the reviesed paper.

Reviewer 2 Report

The authors addressed most of the comments and suggestions performed. However, in the case of the discussion did not mark exactly the changes done. Actually, they mark the complete discussion section what difficult to follow what exactly done.

In the present status the manuscript could be considered for further publication according the integration of the rest of the reviewer comments by the editor.

English should be carefylly reviewed again before publication

Author Response

Response to Reviewer 2 Comments

1. Comments to the Author:However, in the case of the discussion did not mark exactly the changes done. Actually, they mark the complete discussion section what difficult to follow what exactly done.

Response 1: Actually,we have rewritten the whole discussion.  So we highlight the complete discussion section.

2. English should be carefylly reviewed again before publication.

Response 2: Thaknks for the suggestion. we have check the manuscript for the english carefully.